# Development, Management and Utilization of a Kiwifruit (*Actinidia* spp.) In Vitro Collection: A New Zealand Perspective

**DOI:** 10.3390/plants12102009

**Published:** 2023-05-17

**Authors:** Jayanthi Nadarajan, Azadeh Esfandiari, Liya Mathew, Jasmine Divinagracia, Claudia Wiedow, Ed Morgan

**Affiliations:** Food Industry Science Centre, The New Zealand Institute for Plant and Food Research Limited, Palmerston North 4410, New Zealand

**Keywords:** *Actinidia* spp., *Pseudomonas syringae actinidiae*, tissue culture, medium term in vitro storage, germplasm conservation, cryopreservation, biosecurity and compliance

## Abstract

The New Zealand Institute for Plant and Food Research Limited (PFR) supports a large kiwifruit breeding program that includes more than twenty *Actinidia* species. Almost all the kiwifruit accessions are held as field collections across a range of locations, though not all plants are at multiple locations. An in vitro collection of kiwifruit in New Zealand was established upon the arrival of *Pseudomonas syringae* pv. *Actinadiae*-biovar 3 in 2010. The value of an in vitro collection has been emphasized by restrictions on importation of new plants into New Zealand and increasing awareness of the array of biotic and abiotic threats to field collections. The PFR in vitro collection currently holds about 450 genotypes from various species, mostly *A. chinensis* var. *chinensis* and *A. chinensis* var. *deliciosa*. These collections and the in vitro facilities are used for germplasm conservation, identification of disease-free plants, reference collections and making plants available to users. Management of such a diverse collection requires appropriate protocols, excellent documentation, training, sample tracking and databasing and true-to-type testing, as well as specialized facilities and resources. This review also discusses the New Zealand biosecurity and compliance regime governing kiwifruit plant movement, and how protocols employed by the facility aid the movement of pathogen-free plants within and from New Zealand.

## 1. Introduction to New Zealand Kiwifruit In Vitro Collection

### 1.1. History of Kiwifruit In Vitro Collection in New Zealand

Cultivars of the genus *Actinidia* (kiwifruit) are an important export (NZD 4 billion) for New Zealand, resulting in significant investment in breeding programs to ensure the industry remains competitive [1]. Breeding draws on the diversity of a species or genus, requiring access to large numbers of individuals that are valued for the traits they offer. The New Zealand Institute for Plant and Food Research Limited (PFR) supports The Kiwifruit Breeding Centre Limited (KBC) in a large kiwifruit breeding program that includes more than 20 species; while thousands of genotypes of these species are maintained in the field, around 450 genotypes are held in vitro with the germplasm collection comprising a subset of these plants.

The arrival of *Pseudomonas syringae* pv. *Actinidiae*-biovar 3 (Psa-3) (previously known as Psa-V) in the North Island of New Zealand in 2010 [2] had a devastating effect on the health of field collections. More than 80% of diploid and about half of the tetraploid *A. chinensis* plants growing in the PFR research orchard in Te Puke were removed because of Psa-3 infection by 2013 [3]. This prompted the urgent establishment of an in vitro repository so that plants could be held in an environment in which they were less exposed to biotic threats. The in vitro repository is located in Palmerston North, away from field-based collections to reduce risks that might arise through natural disasters (Figure 1a). The value of the collection has been accentuated by the expensive and long quarantine procedures for importation, and because of difficulties in accessing material from some countries of origin. With field grown germplasm under constant threat from biotic and abiotic factors, and the challenges with importing planting material, we need to ensure the conservation of New Zealand’s existing kiwifruit genetic resources. Ex situ conservation using in vitro technologies, including cryopreservation, provides an alternative and complementary storage method for this germplasm. However, PFR’s role is also to ensure our plant material is available to users. Hence, the in vitro collection needs to be fit for purpose for PFR’s and our collaborator’s objectives.

Challenges associated with in vitro collections include establishment, propagation and maintenance, as well as re-establishment of plants to the in vivo environment. Consideration must be given to the health status of plants, as the Psa-3 bacterium has been shown to persist within inoculated in vitro growing kiwifruit plants through multiple culture cycles [5]. Additional key elements that will make the germplasm collection successful are accurate curation, tracking and management of the plants.

### 1.2. Restricted Kiwifruit Plant Material Handling and Movement Limitations

In addition to plants being conserved and curated, they may require steps to remove or verify the absence of pathogens so the plants can be moved (within and beyond New Zealand). Additionally, plants may be held because they host particular pathogens that cannot otherwise be maintained. Following the arrival of Psa-3 in New Zealand, regulations were put in place to restrict the movement of kiwifruit plants due to the uneven distribution of the disease within growing areas. While some regions remain Psa-3-free, others have widespread infections, resulting in strict control of plant movements by New Zealand Kiwifruit Vine Health (KVH) and the New Zealand Ministry for Primary Industries (MPI). KVH, established in 2010 to lead the industry response to the Psa-3 incursion, has been responsible for managing biosecurity readiness, response and operations for the kiwifruit industry since 2012, in conjunction with MPI. Plant material can be moved relatively easily from exclusion to recovery regions (Figure 1b) where Psa-3 is widespread. However, moving plants from recovery to exclusion regions requires extensive testing and quarantine procedures. Although these measures have successfully protected growers in Psa-3-free regions, they resulted in limited access to new cultivars for growers in these regions. To move plants from a recovery region to an exclusion region, they must follow a pathway mandated and approved by both KVH and MPI. In New Zealand, handling of restricted kiwifruit material requires adherence to rules regulating Physical Containment (PC2) facilities. The kiwifruit PC2 laboratory in Palmerston North, operated by PFR since 2011, is used to process field-sourced and potentially Psa-3-infected shoots for initiating in vitro cultures. In 2018, MPI finalized the reopening of the *Actinidia* Import Health Standard (IHS), allowing the importation of kiwifruit plant material for the first time since the Psa-3 incursion in 2010.

This review will focus on development, management and utilization of the PFR-maintained kiwifruit in vitro collection, application of relevant in vitro technologies, and the challenges associated with these activities.

## 2. Use of the In Vitro Kiwifruit Collection for Germplasm Conservation

### 2.1. Establishing and Maintaining the In Vitro Kiwifruit Collection

#### 2.1.1. Plant Initiation in Tissue Culture

Initiation of field-sourced plant materials into tissue culture is a vital step for establishment of an in vitro collection. Efficient establishment of in vitro cultures is affected by genotype, health of the field-grown mother plant, preparation of explant, surface sterilization procedures, medium composition and growth conditions. For our collection, these have all been improved over the years, with many factors found to contribute to initiation success. When the in vitro system was first being established at PFR (and in the context of trying to rapidly rescue many different genotypes from field collections), budwood explants for initiation were obtained directly from the field. Problems reported in subsequent cultures included: high contamination rates and growth problems, such as apical shoot collapse or slow growth with a success rate of c. 10% [6]. The initiation protocol was later improved by sprouting new shoots from canes excised from winter-dormant mother plants in a “clean” indoor environment [7] and modifying the surface sterilization protocol [8]. The original medium onto which the explants were initiated was based on Murashige and Skoog (MS) salts [9] and supplemented with zeatin. Although good for some genotypes, others, particularly males, were reported to stop growing and die after developing symptoms of shoot-tip browning, shoot dieback, leaf necrosis and significant callus growth [6]. To accommodate a wider genotype range, zeatin was replaced with meta-Topolin (mT) in the initiation and proliferation medium comprising half-strength MS macro-elements, full-strength MS microelements, B5 vitamins [10] and 3% (*w*/*v*) sucrose, supplemented with 0.66 mg/L mT, 0.05 mg/L Indole-3-butyric acid (IBA), 0.1 mg/L gibberellic acid and 0.75% (*w*/*v*) agar [11]. These and other modifications in our initiation protocol have led to a success rate of over 70% in establishing in vitro cultures of a wide range of genotypes from different species, this includes removal of plants shown to carry endogenous organisms.

After successful initiation of the explants (growing from axillary buds) in mT medium, young plants are transferred to proliferation medium for six weeks, after which these plants undergo a bacteriological screening (described in Section 2.1.2 below). Any plants exhibiting bacteriological or fungal contamination are discarded, and this is noted in the Germplasm Management System (GMS). Once the plants have cleared bacteriological screening, they are maintained on proliferation medium. A 4–6-week culture cycle is used. The plants are grown under standard culture conditions of a temperature of 24 ± 1 °C, 16 h photoperiod and a light intensity of 35–45 μmol m^−2^ s^−1^, provided by Philips^®^ cool white fluorescent tubes (Philips, China). To induce the growth of roots, healthy plantlets are transferred onto a medium that comprises half-strength MS macro-elements, full-strength MS microelements, Linsmaier and Skoog (LS) [12] vitamins, 3% (*w*/*v*) sucrose, 0.6 mg/L IBA and 0.75% (*w*/*v*) agar. The duration needed for root initiation varies based on the genotype, generally ranging from two to four weeks.

#### 2.1.2. Bacteriological Screening

The importance of plant health status with particular reference to Psa-3 has been mentioned previously. Many of the in vitro cultures were initiated from plants growing in the field. However, even apparently healthy-looking plants have been demonstrated to carry the Psa-3 bacterium [13,14]. Additionally, plants carry a microbiome that reflects their cultivation history and they may harbor endogenous pathogens that cannot be detected using standard plant tissue culture media [15]. This risk is minimized by screening for contaminating organisms following the onset of growth on the tissue culture-initiated explants. For this, the basal part of the plant is excised and transferred into a Petri plate containing a medium comprising 20 g/L potato dextrose agar (PDA) with 5 g/L peptone for one week to encourage proliferation of endogenous pathogens. Plants that show no contamination are retained, but then, depending on the purpose of the in vitro cultures, they might be subjected to further screening for Psa-3 using polymerase chain reaction (PCR) testing in conjunction with incorporation of 3 g/L peptone into growing medium to confirm the plants are Psa-3-free [13,16].

#### 2.1.3. Optimization of Tissue Culture Medium

Perhaps the most significant challenge at present is how to cope with the differences in in vitro performance of the diverse range of kiwifruit genotypes. This is an issue for all germplasm collections that will invariably lack the resources to optimize conditions for each of the large numbers of genotypes they hold. It is generally accepted that genotypes of the same species can respond quite differently to the same in vitro conditions. The same is true for kiwifruit. Although many genotypes will grow in vitro on the same medium and in the same conditions [17,18], different genotypes, even siblings, may have different requirements for optimum in vitro growth. For this discussion, we will focus only on nutrient salt formulations for kiwifruit in vitro culture.

A range of media have been used to grow kiwifruit in in vitro culture since the publication by Harada [19]; however, there has been little optimization of the nutrient medium. Though MS salts are frequently used, other formulations have been used in different applications [17]. In trying to improve establishment of in vitro cultures, Debenham et al. [6] tested genotypes of *A. chinensis* and *A. polygama* on Long and Preece medium (LP) [20], Woody Plant Medium (WPM) [21] and B5 and MS macronutrient salt formulations. Though differences in plant performance were noted, e.g., smaller and paler leaves on LP and B5 media, these were not considered limiting to culture initiation success rates [6]. Half-strength MS medium was also recommended by some research groups [18,22], which we use frequently for our kiwifruit cultures now.

Work to identify an improved salt formulation for in vitro culture of *A. arguta* has been reported. For example, Hameg et al. [23] evaluated the response of an *A. arguta* genotype to six different nutrient formulations. They reported that the medium of Standardi [24] gave the best or equal best results and B5 the poorest results, though they noted that Standardi [24] medium resulted in some undesirable callus growth. A “Design of Experiments” [25] approach was used to formulate a new nutrient salt combination (R medium) for an *A. arguta* genotype [26]. We have begun to evaluate the “R” medium.

### 2.2. Current Kiwifruit In Vitro Collection Status

The PFR kiwifruit in vitro collection currently features more than 450 genotypes from various species. During the four-year period from 2019 to 2023, we have successfully initiated 370 genotypes of kiwifruit into in vitro culture. These genotypes represent *A. chinensis* var. *chinensis* (54%), *A. chinensis* var. *deliciosa* (19%), *A. melanandra* (1%), *A. arguta* (2%) species and hybrids (12%). Additionally, 31% and 49% of the plants initiated are male and female, respectively (Figure 2a,b). In addition to its diverse range of species and genotypes, this collection also preserves genotypes that no longer exist in the field, potentially adding further value to the collections.

### 2.3. Medium-Term Storage

Following the establishment of cultures, plants may be moved to medium-term storage (MTS) for conservation in a system that requires less maintenance of plants. Under the MTS protocol, plants are held in vials of half-strength MS medium supplemented with 3% sucrose and solidified using 0.75% agar under conditions of low temperature (5 °C) and low light (1–3 μmol m^−2^ s^−1^, 8 h photoperiod). MTS methods reduce plant metabolic activity through physical, chemical, or nutrient limitations to plant growth [27]. This approach can reduce the frequency of subculture events from monthly to yearly, providing savings in labor and lowering risks of handling errors and genetic instability. Plants stored under MTS conditions are easily available for propagation or other purposes.

Kiwifruit accessions under MTS conditions undergo two distinct stages, i.e., slow growth storage and rejuvenation. Sixteen healthy plants of a genotype are placed in eight clear screw-top vials with growth medium described above. They are grown in standard culture conditions for four weeks, then undergo a 7-day acclimation period at 22 °C for 10 h/2 °C dark for 14 h. Following acclimation, these plants are moved to the MTS environment as described above. Visual evaluations of plant quality / health are conducted once every three months. When only 20% of the plants remain viable, they are subjected to a rejuvenation step. To accomplish this, plants are removed from all eight vials and undergo two or three rounds of subcultures to generate a new set of 16 plants, which are then returned to the MTS environment. The proliferation process begins with a medium containing mT and ends with a medium containing zeatin [6], resulting in a larger number of plants in a short period of time with a standard plant size of 3 to 4 cm for MTS. The duration of a plant can be maintained under MTS before rejuvenation is species and genotype dependent. As an example, the storage life of four genotypes of *A. macrosperma* and *A. chinensis* at 5 °C are presented in Table 1. In general, plants can be held at MTS conditions without rejuvenation for between 400 and 800 days. A small proportion (about 4%) require rejuvenation within 200 days, while a low proportion can last between 1200 and 1400 days (Figure 3). All activities are recorded in a database where the vials containing plants are labelled with details of the genotypes, vial identity, and date. All plant transfers are recorded using our in-house database systems, i.e., Germplasm Management System (GMS) and Vial Management System (VMS) (Figure 4). Trays are used to store the vials and are assigned a number in VMS to easily locate the genotypes.

### 2.4. Long-Term Conservation Using Cryopreservation

The in vitro plants provide starting material for long-term conservation using cryopreservation. Cryopreservation—the preservation of viable cells, tissues, organs and organisms at ultra-low temperatures (c. −196 °C) in the liquid or vapor (c. −150 °C) phase of liquid nitrogen. This method of preservation is growing in popularity for plant germplasm conservation due to its comparatively low maintenance cost, small space requirement and reliability [28]. Other advantages of cryopreservation over medium-term storage (MTS) or standard in vitro storage are the reduced threats of contamination and somaclonal variation [29].

Cryopreservation protocols are usually developed empirically for specific materials or explants, taking into consideration the physiological and bio-physical factors of the explants to minimize stress and maximize survival rates [30]. Cryoprotection can be applied in the form of osmo-protection alone, i.e., direct desiccation using silica gel or air drying, or in a combination of osmo-protection and chemical cryoprotection using permeating chemical cryoprotectants. The ultimate goal is to achieve a ‘vitrified state’ which is the critical physical state that determines the success of the post-cryopreservation survival [31]. The development of a simple and reliable cryopreservation protocol would allow wider application of this preservation technique in conservation of plant materials.

At PFR, a cryopreservation protocol has been developed for the long-term storage of kiwifruit using a droplet vitrification protocol [18]. This protocol was successfully tested on nine genotypes from five species, i.e., *A. chinensis* var. *chinensis*, *A. chinensis* var. *deliciosa*, *A. arguta*, *A. macrosperma* and *A. polygama*, utilizing shoot tips around 1 mm in size excised from two-week-old axillary shoots (Figure 5). These young shoots were obtained by growing nodal cuttings sourced from six-week-old in vitro grown mother plants. In our study, we observed that the age of the mother plant and size of the explant can determine the success of cryopreservation; use of younger donor plants and smaller explants significantly increasing regeneration [18]. The response of kiwifruit plants to in vitro culture is not just species-specific but also genotype-specific [32]. Hence, significant variation in post-cryopreservation regeneration between different genotypes was expected, and was observed to range from 59% to 88% [33,34]. Our results confirmed that different kiwifruit genotypes respond differently to the stresses imposed at various stages of the cryopreservation protocol [33,34]. Since post-cryopreservation regeneration is higher than the suggested baseline of 40% for cryobanks [35], we will be implementing this cryopreservation protocol for long-term kiwifruit germplasm conservation.

## 3. Use of In Vitro Collection for Plant Movements

The key role of the collection and collection facility is to ensure the availability of plants for breeding and research programs and to support industry partners. The laboratory provides a hub for distributing tissue-cultured plants to a range of users in New Zealand and offshore. A goal is to ensure supply of Psa-3 free plants. Any such movements of plant material are subject to appropriate consent/material transfer agreement from the owners of these genotypes. These plant movements comply with previously mentioned requirements from MPI and KVH, where PFR is tracking all exported plants in the GMS database (date, genotype, number of clones, etc.) audited six-monthly or annually by MPI and KVH.

### 3.1. Plant Movement Overseas (Export)

Before plant material leaves the facility, a true-to-type quality control assessment is carried out using DNA fingerprinting. Molecular techniques are commonly used in plant breeding, intellectual property (IP) protection and cultivar verification, and to facilitate plant genetic resources management and conservation [36]. Standard lab protocols have been adjusted and developed at PFR over the past few years for DNA profiling and are described in Rowland et al. [37] and Knäbel et al. [38], using a modified cetyl-trimethylammonium bromide (CTAB) method [39,40] and in-house developed simple sequence repeat (SSR) markers that are species-specific for kiwifruit [41].

Using SSR, amplified fragments of DNA are scored as illustrated in Figure 6, where DNA fragments and alleles are displayed as peaks with their sizes in base pairs (bp). The resulting molecular characterization/scoring output shows patterns of molecular similarity indicating if the studied plant is clonal to a reference sample as shown in Figure 6.

Following true-to-type testing, plants that have passed both phases (bacteriological and DNA profiling) are prepared for shipment and export following packaging one clonal plant per vial, and labelled with a GMS barcode to provide a unique identifier. Typically, 10–12 vials of rooted plants will be provided per genotype. Vials are triple-packed and sent according to IATA shipping standard 650 [42]. Figure 7 summarizes the flow of activities (corresponding to GMS activities) for overseas exports. For example, when a plant is received, its details are placed into the GMS database and it undergoes a series of activities. These activities correlate to specific GMS activities that allow for tracking plants in specific stages. Activities for export of plants are highlighted in violet (DNA fingerprinting, peptone testing, rooting and export). Discarded accessions are also recorded in GMS (indicated in pink). The number of peptone tests carried out corresponds to the specific movement pathway. For example, for plants being sent within New Zealand, a PCR test and a minimum of three peptone tests are required (Figure 7).

### 3.2. Plant Movement within New Zealand

It is sometimes desirable to move plant material bred in a recovery region (North Island) to an exclusion region (South Island) (Figure 1) to ensure commercial growers are accessing new cultivars (and to support breeding activities in other regions). In this instance, the plants go through periods of greenhouse and outdoor growing quarantine before being released into exclusion regions (Figure 1) [43]. Candidate plants must be stage 3 tissue culture plants (rooted plants) of confirmed Psa-3-free status. This is achieved by identifying Psa-3-free in vitro grown mother plants from which further plants are propagated. Psa-3-free status is first indicated by PCR testing of individual plants. These plants are subsequently propagated, with progeny undergoing repeated screenings (at least three times) on peptone-supplemented medium. If any plant derived from a mother plant is subsequently found to have Psa-3, all plants of the lineage are required to be destroyed. These steps are intended to provide industry and regulatory authorities a high degree of confidence that the in vitro plants have minimal chance of carrying Psa-3 and can be moved from containment to a Psa-3 quarantine greenhouse within the South Island. Plants are held in a Psa-3 quarantine facility for up to two years before release.

In addition to the South Island pathway, there is also a pathway for the movement of candidate plants within North Island recovery regions. The candidate plants undergo PCR screening to identify disease-free mother plants followed by subsequent screening on a peptone-supplemented medium to ensure Psa-3-free status. After achieving “in vitro,” Psa-3-free status candidate plants may, with prior approval from KVH and MPI, be transferred from a containment facility to a non-containment facility.

### 3.3. Plant Movement into New Zealand (Import)

The significant loss caused by the Psa-3 incursion has resulted in New Zealand adopting a very cautious approach to new importations of *Actinidia*. All plant material must go through a tissue culture step or be imported as tissue cultures. If dormant cuttings are imported, they must be used to generate tissue cultures in a quarantine facility [44,45]. Although the current facility is not involved in imports of kiwifruit plants into New Zealand, this point is highlighted, since the requirement for the importation pathway mandates in vitro steps which places further pressure on the development of improved in vitro protocols.

## 4. Use of the In Vitro Collection to Maintain a Virus Reference Collection

There is value in maintaining reference collections, not just of plants but also of their pathogens. In some cases, the host range of the pathogen may be limited, or the pathogen may not be widely dispersed in the environment, and it may be convenient to maintain infected in vitro cultures. This is a method we use to maintain access to the betaflex virus, *Betaflexiviridae*, with a small number of different kiwifruit genotypes known to be infected held in culture.

## 5. Management and Challenges Associated with In Vitro Collection

Since 2014, all data from the PFR field kiwifruit germplasm and breeding collections have been held in Ebrida, a plant-breeding software package developed by Agri Information Partners (Wageningen, The Netherlands) and adapted for use in PFR (www.e-brida.nl, accessed on 2 February 2023). This database contains information on accessions with their taxon, passport data and ploidy, block plans, listing individual vines with their unique ID(s), planting position(s), year planted, sex, phenotypic and/or genotypic data. The Ebrida database ensures kiwifruit germplasm and breeding information is available to approved users of the collection, including breeders, germplasm curators, biometricians and molecular geneticists.

### 5.1. Database and Sample Tracking Systems

For management and sample tracking of the in vitro collection, GMS is used. The GMS is a user-friendly, searchable database designed to track and record all details of genotypes and plants in the in vitro collection. Plants are tracked from initiation (including passport data) to disposal with all details recorded in the database. A discarded sample is also recorded in the system, so it can be traced to determine if a genotype still exists in the collection. The system assigns unique “Activity Event ID” codes to each container and can link each container back to its previous “Activity Event ID”, allowing for a complete history of events related to each plant to be recorded, right back to the plant in the field [46] (Appendix A). One important component of the GMS is the VMS, which is specifically designed to store and organize data related to genotypes conserved in MTS. The VMS is used to maintain accurate and up-to-date records of all vials and genotypes in MTS, including information on the availability, the location of a vial and quality assessments of the conserved samples. Having accurate genotype information and ensuring the availability of all data upon sample receipt has been crucial for establishing a functional database. It is crucial to establish, follow and audit systems to ensure the best practices for sample collection, storage and analysis are practiced.

### 5.2. Somaclonal Variation

Somaclonal variation can be detected at morphological, cytological, cytochemical, biochemical, and molecular levels [47]. In in vitro cultures kiwifruit, the analysis of genetic stability over the long-term especially in slow growth storage is still lacking. Both genotypic and phenotypic assessments are needed to confirm if somaclonal variation is indeed a problem. Though extreme levels of somaclonal variation between 100% [48] and 90% in banana [49] have been reported, typically an average of 15–20% can be expected [50], depending on the number of culture cycles and culture conditions. The levels of somaclonal variation can vary with genotype, type of tissue, explant source, medium components and the duration of the culture cycle [51]. Some level of somaclonal variation may be inevitable in tissue culture; however, manipulating, controlling and minimizing the putative inductive factors will reduce the level of occurrence. The PFR kiwifruit in vitro collection is being studied for somaclonal variation. We are examining morphological changes in a subset of frequently subcultured genotypes from our MTS collection and using DNA sequencing to detect somaclonal variation at the molecular level.

### 5.3. Backing up Collections

To mitigate the risk of losing the in vitro collection due to environmental catastrophes or human errors, duplication or back up of the collection is recommended [52]. The backup collection should be in a different location to reduce risks to the collection from natural disasters such as earthquake, flood or fire. For our kiwifruit in vitro collection, we are planning to back up the collections in MTS and in cryostorage at different PFR sites. These sites are selected to provide the same storage environment as the original collections. Our database, particularly VMS, has been upgraded to capture the information on backup collections. The backup repository is only responsible for receiving and storing the plant materials and will not be responsible for managing the collection, as all preparation of material (initiation, proliferation, rejuvenation) and database entry will be carried out in our original repository.

### 5.4. Specialized Operations and Resources Requirements for In Vitro Collection

Setting up an in vitro repository requires tissue culture laboratory, medium preparation room, growth chambers, controlled cabinets for medium term storage, cryopreservation facility and an efficient inventory system. Management of an in vitro facility entails cost, particularly expenses associated with electricity, culture medium and labor charges, in addition to the equipment cost. The cost of electricity, which is mainly utilized for autoclaving, artificial lighting in tissue culture rooms, air filtration of laminar air hoods and air conditioning can amount up to 60% of tissue culture production costs [53]. Culture medium is indispensable to a tissue culture facility, but preparation can be expensive. The cost of labor accounts for 60–70% of the medium cost in a tissue culture laboratory [54].

In addition to the infrastructure requirement, highly skilled staff are the backbone of an in vitro collection facility. The staff are trained in aseptic technique, require knowledge of how to modify protocols and must have manual dexterity with aseptic tools, especially when handling fragile plant tissues. Micropropagation is labor-intensive work [54,55]. Since tissue culture media are high in nutrients and plants are often held in a warm environment, microbial contamination can result in significant losses. Prevention of microbial contamination is crucial in micropropagation facilities [56] and requires highly skilled technicians. Ahloowalia and Savangikar [57] and Datta [58] pointed out that using a highly skilled work force can significantly increase laboratory efficiency and reduce costs.

## 6. Conclusions

In this review, we reported the ongoing development of the kiwifruit in vitro collection held at PFR. The in vitro collection complements field collections as repositories for New Zealand kiwifruit genetic resources. The large in vitro collection needs careful curation and management procedures to address the challenges posed by the genetic diversity it holds. The facilities and staff also support end users of plant material by facilitating plant movements within and from New Zealand. The facility provides Psa-3-free plants, reference collections and plants used as resources for research and new cultivar development. The specialized facilities, resources and skilled staff are key to the ongoing management and growth of this collection. We also discussed the need to work with regulators who manage New Zealand’s biosecurity and compliance requirements to ensure plants can be made available to users.

## Figures and Tables

**Figure 1 plants-12-02009-f001:**
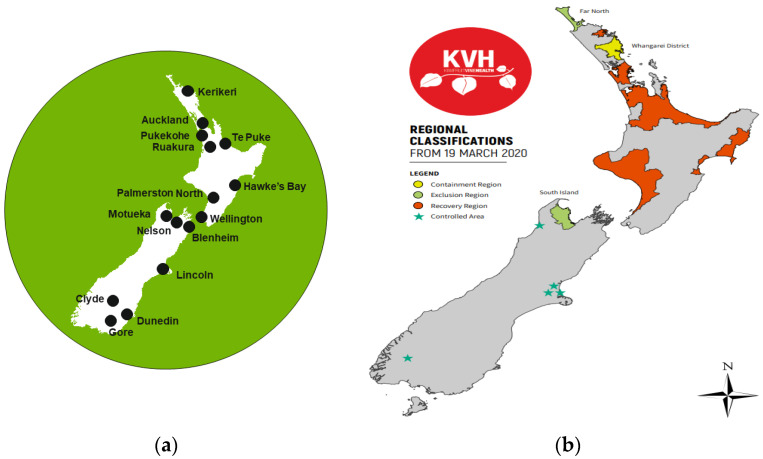
(**a**) Plant & Food Research (PFR) locations within New Zealand. PFR kiwifruit collection is grown in orchard blocks at Kerikeri, Ruakura, Te Puke, Motueka and Clyde; (**b**) Map showing the boundaries and status of regions in respect of the presence of *Pseudomonas syringae* pv. *Actinidiae*-biovar 3 (Psa-3) (adapted with permission from KVH [4]). Containment region (yellow): Limited Psa-3 infections. Movement of plant material is restricted; Exclusion region (green): No orchards with Psa-3 identified, or Psa-3 has recently been identified for the first time. Material is unrestricted; Recovery region (red): Psa-3 is already widespread. Material is restricted; Controlled areas (stars): only within the South Island. Exclusion regions and movement restrictions apply.

**Figure 2 plants-12-02009-f002:**
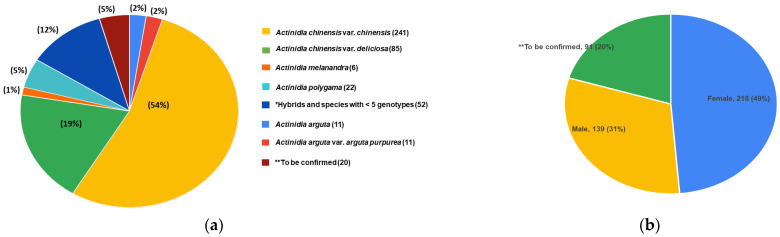
The diversity of the genotypes in the PFR kiwifruit in vitro collection. (**a**) Number and per-centage of genotypes for each species. (**b**) Genotypic distribution by sex in number and percentage. * This collection includes one genotype each from *A. callosa*, *A. chrysantha*, *A. chinensis* var. *deliciosa coloris*, *A. latifolia* and *A. valvata*; two genotypes each from *A. hemsleyana*, *A. indochinensis* and *A. setosa*; three genotypes of *A. kolomikta;* and five genotypes from *A. macrosperma* species, respectively. ** Gap analyses are underway to confirm the species and sex of the unidentified accessions.

**Figure 3 plants-12-02009-f003:**
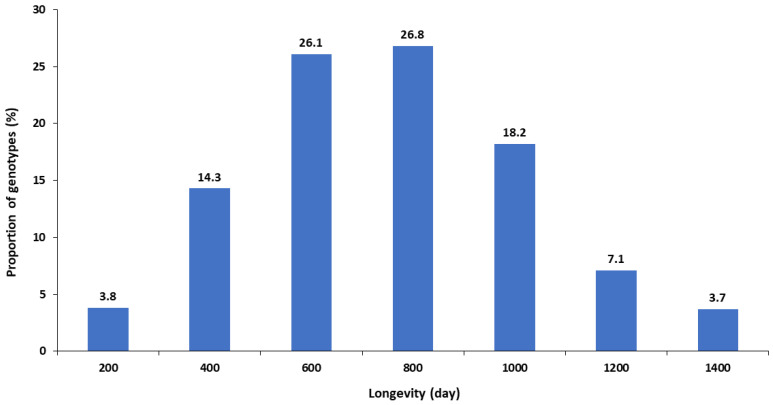
The proportions of kiwifruit genotypes (%) that could be maintained in medium term storage (MTS) for different periods without rejuvenation. MTS is in vitro storage of kiwifruit plants at 5 °C with longevity describing the length of time in days that the plants could be left before requiring rejuvenation.

**Figure 4 plants-12-02009-f004:**
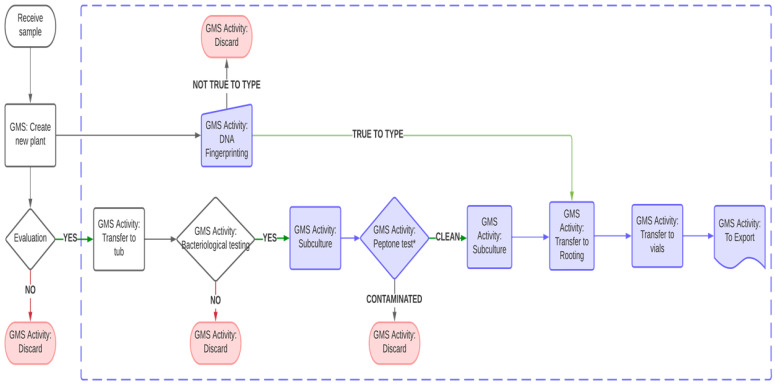
Flow diagram summarizing steps involved in placing a plant into medium-term storage (MTS) and tracking of plants using Germplasm Management System (GMS) and Vial Management System (VMS). When a plant is received, its details are entered into the GMS database and the plant is prepared for MTS at 5 °C. GMS provides details of every plant in the laboratory with VMS used to record plants being held in vials in MTS. The activities highlighted in blue are activities recorded in VMS. Discarded accessions are also recorded in the system (indicated in pink).

**Figure 5 plants-12-02009-f005:**
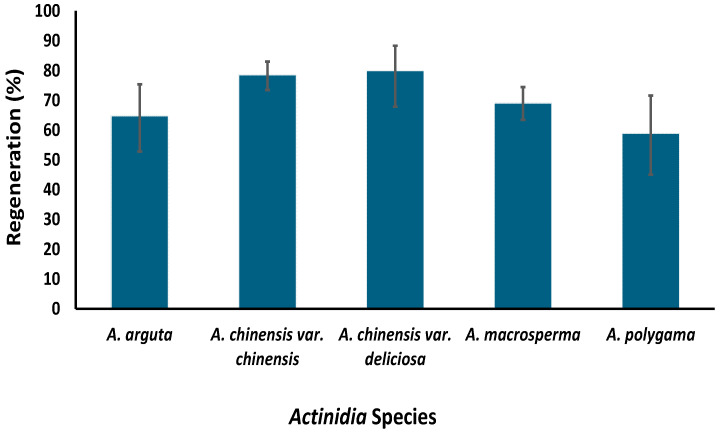
Regeneration percentages of five kiwifruit species following cryopreservation using a protocol established at The New Zealand Institute for Plant and Food Research Limited (reproduced from Table 2 [18] with permission from Springer Nature License Number 6013930561362).

**Figure 6 plants-12-02009-f006:**
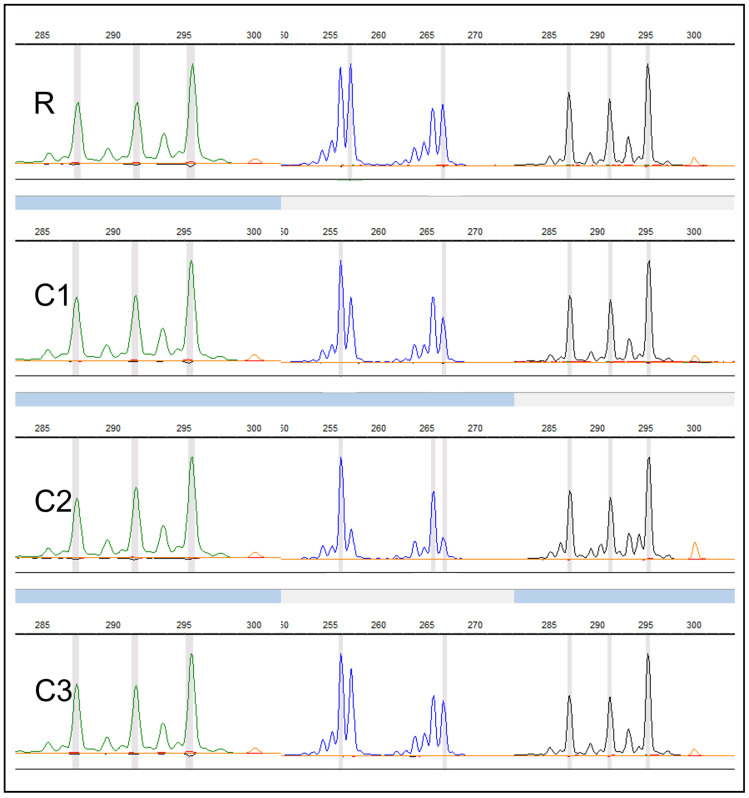
SSR-amplified fragments of a kiwifruit genotype. An example output of GeneMarker^®^ v 2.2.0 software (© SoftGenetics, LLC, State College, PA, USA) showing R = reference sample displaying identical allele/peak pattern as clone 1, 2 and 3 (C1–3) and suggest true to type.

**Figure 7 plants-12-02009-f007:**
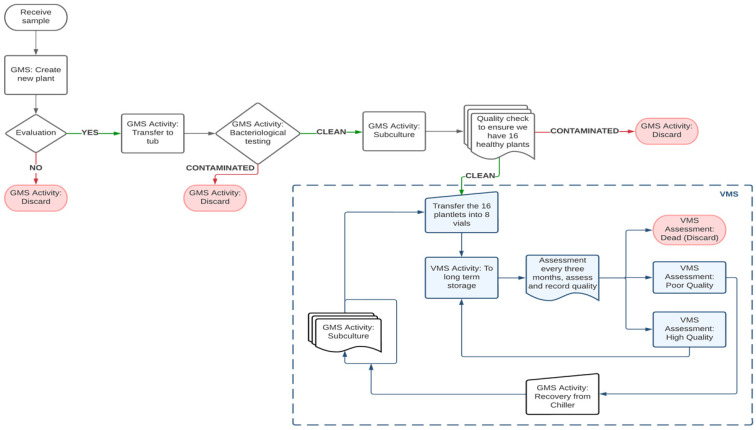
Summarized workflow for preparation of in vitro plants for overseas trials and plant movement within New Zealand at the PFR in vitro facility.

**Table 1 plants-12-02009-t001:** Selected examples of genotypes in medium-term storage (MTS). Genotypes 11384 and 6001 are male plants of *Actinidia macrosperma* (MA) and genotypes 11348 and 11158 are female plants of *Actinidia chinensis* var *chinensis* (CK). These genotypes have undergone repeated cycles of storage and rejuvenation, with varying storage duration at 5 °C.

Genotype ID	Species	Gender	Year of Entry	Average of Days Being in MTS
11384	MA	Male	2019	955
6001	MA	Male	2018	496
11348	CK	Female	2017	349
11158	CK	Female	2016	680

## Data Availability

The data presented in this study are available on request from the corresponding author.

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
