# Peer review of "Development, Management and Utilization of a Kiwifruit (Actinidia spp.) In Vitro Collection: A New Zealand Perspective"

_plants, 2023, doi:10.3390/plants12102009_

Round 1
Reviewer 1 Report
Dear authors,
I can congratulate for this very nice and valuable work. I did not find any mistake and the design and all parameters are well explained.
I recommend it to be publish in a present form.
Regards
Zvjezdana
Author Response
We are grateful for the positive comments and appreciation of our work.
Kind regards,
J Nadarajan et al
Reviewer 2 Report
The present submission reviewed the work done by the New Zealand Institute to facilitate the development and management of kiwifruit genetic resources. This review will be beneficial to Genebank managers to implement integrated conservation systems for the safe conservation and utilization of plant genetic resources. Please find in the attachment some suggestions for minor revisions that should be made before it could be accepted for publication.

Author Response
Authors Response: We appreciate positive feedback on our review. Specific comments are addressed below;
Specific Comments:
- Please consider if the use of ‘utilization’ is appropriate for the introduced contents of this review. From my perspective, the utilization of kiwifruit in vitro collections means the in vitro maintained plants were used in breeding or for other scientific purposes, rather than just for conservation.
Authors Response: Our in vitro collections are indeed used for assisting in breeding programmes, overseas export trials and scientific research activities in addition to conservation purpose. Therefore, we would like to retain the word ‘utilization’ in the title.
- Moved lines 55-58 to Section 1.2. Now lines 73-75.
- Moved lines 74-77 to end of Section 1. Now lines 97-99.
- Line 114: Modifying surface sterilization protocol - This part should be extended to introduce some detailed information regarding the optimized disinfection protocols.
Authors Response: As the modified surface sterilization protocol is described in detail in Ref 8 (Debenham, M. and R. Pathirana, Establishment and management of an in vitro repository of kiwifruit (Actinidia spp.) germplasm, in Meta-Topolin: A Growth Regulator for Plant Biotechnology and Agriculture. 2021, Springer. p. 279-291), we feel there is no need to repeat the same information here.
- Section 2.4. is now moved 2.2; 2.2 to 2.3 and 2.3 to 2.4. All figures are renumbered accordingly.
- Figure 5: Figure marks too small; the original figure supplied has larger legends.
- Line 197: ‘cool storage’ is now changed to ‘slow-growth storage’ (now line 206-207)
- Line 238: added ‘–‘ between ‘Cryopreservation’ and ‘the’ (now line 247)
- Line 262: ‘Kiwifruit plants response’ is changed to ‘The response of kiwifruit plants’ (now line 273)
- Line 248: It may be better to highlight the vitrification solution-based methods for cryoprotection which combines the osmo-protection using high level of sugars with chemical dehydration by plant vitrification solutions.
Authors Response: Here we are highlighting on achieving a ‘vitrified’ state by just osmo-protection using desiccation (laminar air flow cabinet or silica gel) or combination of both osmo-protection and chemical-protection using sucrose and permeating cryo-protectants such as DMSO. We have expended the sentence to emphasis these points (lines 256-261).
- Line 320: ‘providing’ changed to ‘to provide’ (now line 315)
Reviewer 3 Report
The manuscript is a review of the development and use of in vitro collections of Kiwifruit in New Zealand.
A detailed description is given of each of the steps involved in establishing and maintaining collections for germplasm conservation.
Regarding genotypes, are they all diploid? Figure 6, SSR markers, shows more than two alleles per locus.
Best regards
Author Response
Authors Response for Figure 6: No, we have a range of ploidies in our kiwifruit collection. The example shown in Figure 6 is not diploid and therefore showing more than two alleles per locus.
Kind regards,
J Nadarajan et al